# Real-World Effectiveness of COVID-19 Vaccines against Severe Outcomes during the Period of Omicron Predominance in Thailand: A Test-Negative Nationwide Case–Control Study

**DOI:** 10.3390/vaccines10122123

**Published:** 2022-12-12

**Authors:** Natthaprang Nittayasoot, Rapeepong Suphanchaimat, Panithee Thammawijaya, Chuleeporn Jiraphongsa, Taweesap Siraprapasiri, Kritchavat Ploddi, Chakkarat Pittayawonganon, Surakameth Mahasirimongkol, Piyanit Tharmaphornpilas

**Affiliations:** 1Division of Epidemiology, Department of Disease Control, Ministry of Public Health, Nonthaburi 11000, Thailand; 2International Health Policy Program, Ministry of Public Health, Nonthaburi 11000, Thailand; 3National Institute of Health, Ministry of Public Health, Nonthaburi 11000, Thailand

**Keywords:** vaccine effectiveness, pneumonia, COVID-19, SARS-CoV-2, Omicron, Thailand

## Abstract

Due to the widespread Omicron variant of SARS-CoV-2 in Thailand, the effectiveness of COVID-19 vaccines has become a major issue. The primary objective of this study is to examine the real-world effectiveness of COVID-19 vaccines based on secondary data acquired under normal circumstances in a real-world setting, to protect against treatment with invasive ventilation of pneumonia during January to April 2022, a period when Omicron was predominant. We conducted a nationwide test-negative case–control study. The case and control were matched with a ratio of 1:4 in terms of age, date of specimen collection, and hospital collection specimen and the odds ratio was calculated using conditional logistic regression. Overall, there was neither a distinction between mix-and-match regimens and homologous mRNA regimens against severe symptoms, nor was there a decline of the protective effect over the study period. The third and fourth dose boosters with ChAdOx1 nCoV-19 or mRNA vaccines provided high levels of protection against severe outcomes, approximately 87% to 100%, whereas two doses provided a moderate degree (70%). Thus, this study concludes that current national vaccine strategies provide favourable protective benefits against the Omicron variant. All Thais should receive at least two doses, while high-risk or vulnerable groups should be administered at least three doses.

## 1. Introduction

As of 5 May 2022, there have been more than 500 million confirmed cases of coronavirus disease 2019 (COVID-19) and over 6 million deaths worldwide since the pandemic began [1]. To control the spread of disease, numerous measures were implemented, including city lockdowns, community and home isolation/quarantine, active case finding in communities, movement restrictions, and mandatory mask use [2]. Among these measures, some experts believed that vaccination is one of the most promising measures, as it is a safe method to establish herd immunity and can reduce the possibility of an infected person experiencing severe illness and death [3,4,5].

Thailand began vaccinating its population on 28 February 2022 [6]. Since then, four vaccines, namely CoronaVac (produced by Sinovac Biotech, Beijing, China), ChAdOx1 nCoV-19 (AstraZeneca, Oxford, UK), BBIBP-CorV (Sinopharm, Shanghai, China), BNT162b2 (Pfizer-BioNTech, New York, NY, USA), and mRNA1273 (Moderna, Cambridge, MA, USA), have been distributed by the government throughout the country [6]. Amid the limited number of vaccines, and to reduce the surge of cases during the period when the Delta variant was a dominant strain, the Thai Government launched a mix-and-match vaccination approach to inoculate its population faster [7]. CoronaVac, an inactivated vaccine, was utilized as the first dose, and ChAdOx1 nCoV-19, a viral vector vaccine, as the second dose, with a 3–4 week interval between doses [8,9]. Later, further combination regimens (e.g., one ChAdOx1 nCoV-19 with one BNT162b2) were introduced [10]. Since January 2022, a third dose and fourth dose with ChAdOx1 nCoV-19, BNT162b2, or mRNA1273 was recommended [11,12]. As a result, Thailand has a number of vaccine regimens that are different from those used in many other countries and some of those regimens have limited evidence of real-world vaccine effectiveness.

Thailand is currently experiencing the country’s fifth wave of the COVID-19 pandemic, with the highly contagious Omicron variant accounting for 70% of all strains in January, and 100% in April 2022 [13]. Since 1 January 2022, the start of the fifth wave, over 2 million confirmed cases and 7216 deaths have been documented [14]. With a continuous increase in reported daily caseloads, the true incidence may be much higher. Vaccination is believed to be a crucial intervention, and its coverage is considered a key determinant in ending the pandemic. Thus, the real-world effectiveness of COVID-19 vaccine regimens implemented in Thailand, particularly during the period when the Omicron variant prevails, is urgently required.

As COVID-19 is going to be an endemic disease, not only in Thailand but also worldwide, this study pays attention to the severe outcomes of the disease rather than overall number of infections. The aim of this study is to determine the real-world effectiveness of COVID-19 vaccine regimens against treatment with invasive ventilation in Thailand.

## 2. Materials and Methods

### 2.1. Study Design

An individually matched test-negative case–control study was conducted with a case–control ratio of 1:4. The study population was Thai individuals tested for SARS-CoV-2 between 1 January and 30 April 2022 by healthcare professionals. The indications for testing included those at high-risk of being in contact with a COVID-19 case and active case finding in communities by outbreak investigators.

A COVID-19 case was classified as an individual who had a positive test for SARS-CoV-2 by reverse transcriptase polymerase chain reaction (RT-PCR) or professional-use antigen test kit (ATK) at the laboratory certified by National Institute of Health. Additionally, the case had no prior history of SARS-CoV-2 detection in the preceding three months. A control was an individual who had not been detected with SARS-CoV-2 by RT-PCR or ATK. Case and control subjects were matched with ratio of 1:4 with respect to age (allowing a three-year margin). The other two matched variables included date of laboratory collection (allowing thirty-day margin) and site of specimen collection.

### 2.2. Data Sources

We analyzed vaccine effectiveness using secondary data, which were retrieved from four main national health databases: Co-Lab (national monitoring system for COVID-19 laboratory tests, operated by Department of Medical Sciences [DMSc]), Co-Ward (national monitoring system for COVID-19 cases admitted in health facilities, operated by Office of Permanent Secretary [OPS]), COVID-19 Death (national monitoring system on deaths, operated by OPS), and MOPH-IC (national immunization information center, operated by Office of Permanent Secretary [OPS]). We used the national identification numbers of each individual as a unique identifier to link the same person across databases.

We used the Co-Lab database to access demographic data of individuals, test results, indications of testing, and laboratory collection sites. The Co-Lab database retrieved laboratory test results from public and private hospitals, provincial public health offices, and regional offices of disease prevention and control.

For assessing the severity of COVID-19 cases, we used information from the Co-Ward database and the COVID-19 death database. The Co-Ward database collects data of COVID-19 cases severity admitted in both public and private health facilities. The database stores information about the severity of COVID-19 cases (asymptomatic, mild, moderate, and severe), medications, bed occupancy, and types of oxygen support. Severe cases were defined as cases with pneumonia-like symptoms and with oxygen saturation of less than 95% in room air [15].

For vaccination status, the MOPH-IC database contains information on all individuals administered with COVID-19 vaccines in Thailand. It is a mandatory health data system; therefore, the likelihood of missing records is limited. The MOPH-IC database collects information about national identification number, risk category, vaccination date, and vaccine manufacturers.

### 2.3. Statistical Analysis

We started with descriptive statistics to examine the overview of the data. Categorical data were presented descriptively using frequency counts and proportions while continuous variables were presented with mean and standard deviation (SD).

For the analysis on vaccine effectiveness (VE), we used conditional logistic regression where the primary outcome was severe COVID-19, characterized as requiring invasive oxygen support. Controls were participants who did not meet the severe definition. The secondary outcome was any infection, defined as individuals who were detected with SARS-CoV-2, by RT-PCR or ATK whereas controls were those showing negative test results. In terms of death, the case was a participant who died of COVID-19 while controls composed of non-SARS-CoV-2 infected individuals and non-dead COVID-19 cases.

Vaccination status served as the independent variable. Participants were regarded as a vaccine recipient if they were administered the vaccine at least fourteen days prior to the sample collection. We excluded participants who had received the last dose of vaccine within fourteen days before the sample collection. Unvaccinated individuals were defined as those who had not been vaccinated prior to laboratory collection.

An odds ratio (OR) was calculated with a 95-percent confidence interval and vaccine effectiveness was defined as one minus the OR multiplied by one hundred. Matching was used to account for potential confounders, namely, age, address and residential area of the hospitals, and time of infection (months of SARS-CoV-2 test).

We assessed the vaccine effectiveness by the number of vaccines administered stratified by age group (18 to 59 years and greater than or equal to 60 years). The time interval between vaccination date and sample collection was also categorized and stratified to explore the vaccine effectiveness over the six-month course. Additionally, we evaluated the effectiveness of two, three, and four doses of COVID-19 vaccine by each regimen.

### 2.4. Ethics

The study was approved by the institutional review board of Thai Department of Disease Control, Ministry of Public Health (Reference number: 65005; FWA number 00013622).

## 3. Results

We obtained a total of 3,059,616 records including: 1015 cases of COVID-19 pneumonia requiring invasive ventilation (0.16%) from 652,854 cases with SARS-CoV-2 detection and 2,406,762 controls or non-SARS-CoV-2 detection, during 1 January to 30 April 2022. As shown in Table 1, the majority of cases and controls were aged between 18 and 59 years, with a mean age of approximately 37 years. Approximately 0.2% of the cases were found to have pneumonia requiring invasive ventilation and 0.1% of the cases died.

Figure 1 shows vaccine effectiveness for the three outcomes stratified by number of doses received. Both single and two-dose vaccinations did not provide protection against SAS-CoV-2 infection. The vaccine effectiveness among individuals who received three doses was 5.91% (95%CI 4.77–7.03%), and among those receiving four and five doses it was 71.11% (95%CI 70.55–71.65%) and 83.13% (95%CI 77.26–87.49%), respectively. The vaccine effectiveness against pneumonia requiring invasive ventilation increased with the number of vaccine doses received. For those who received two doses, the effectiveness was 70.41% (95%CI 64.59–75.27%). For three doses, the effectiveness was 90.39% (95%CI 87.30–92.73%), slightly lower than the effectiveness for four doses, where the estimate rose to 99.59% (95%CI 96.97–99.94%).

Figure 2 shows the vaccine effectiveness against pneumonia requiring invasive ventilation stratified by age group and number of doses received. Overall, the vaccine effectiveness was comparable between age groups. A single dose of COVID-19 vaccine showed effectiveness of about 50%, while two doses resulted in an effectiveness of 75.71% (95%CI 64.61–83.32) and 68.29% (95%CI 60.91–74.27%) for individuals aged 18–59 years and ≥60 years, respectively.

Figure 3 demonstrates the vaccine effectiveness against pneumonia requiring invasive ventilation, stratified by the duration between the last vaccination and sample collection. Effectiveness increased with the number of doses and the pattern was consistent across the three durations. Vaccine effectiveness for those receiving three and four doses ranged from around 80% to 100%.

Table 2 presents the vaccine effectiveness against pneumonia requiring invasive ventilation for two- and three-dose vaccinees tallied by vaccine regimens. Among the two-dose vaccinees, the effectiveness of heterologous regimens was approximately 70% to 80%, comparable to the homologous mRNA vaccines. Among the three-dose vaccinees, all vaccine regimens showed the effectiveness of at least 80%. Using ChAdOx1 nCoV-19 or any mRNA vaccines as a third dose showed a high effectiveness of about 90%.

## 4. Discussion

This study is a large study using nationwide surveillance data. For the real-world effectiveness of overall vaccine combinations, we found that as the dose of vaccines increased, the severity of COVID-19 assessed by oxygen therapy decreased. There is no distinction between mix-and-match regimens and homologous mRNA regimens against severe symptoms and deaths. The study also emphasizes the importance of the third dose and fourth dose boosters with ChAdOx1 nCoV-19 or mRNA vaccines as they can provide substantial protection against pneumonia requiring invasive ventilation and fatalities in adults and the elderly; however, three doses of the COVID-19 vaccine provided minimal protection against COVID-19 infection. As this study was conducted using data obtained during January to April 2022, (which is when the Omicron variant (BA.1 and BA.2 lineage) became a dominant strain in Thailand from more than 80% coverage in January to 100% in April) our findings conclude that the current national vaccine strategies provide favourable protective benefits against the Omicron variant [13].

Previous evaluations of three-dose vaccine effectiveness against SARS-CoV-2 infection in Thailand, when the Delta variant assumed the dominant share, reported the effectiveness level against SARS-CoV-2 infection at around 80–90% for three doses [16,17,18]. This is considerably greater than the levels observed in our study concerning the Omicron variant. Our results are similar with studies by Hung Fu Tsung and Chariyalertsak who reported the reduction in vaccine effectiveness against the infection of Omicron variant compared to the Delta variant [18,19]; however, the point estimate of effectiveness in our study is lower than Hung Fu Tsung’s study and Chariyalertsak’s study. It might be explained by the different vaccine regimens, as Hung Fu Tsung’s study focused only those receiving the mRNA vaccine. Chariyalertsak’s study included the sample with a valid history of exposing SARS-CoV-2 from outbreak investigations. Compared with the neutralizing antibody study by Cheng and his team, this study showed different results [20]. Cheng’s study suggested that the three doses of COVID-19 vaccine increased neutralizing antibody levels after vaccine administration [20]. This might be caused by the difference in the time period of interest, as Cheng’s study employed shorter study duration after the third doses. As our study estimated real-world vaccine effectiveness while Cheng’s study focused on the neutralizing antibody level, the result may be dissimilar and incomparable. Our study relied on secondary data, which might include non-SARS-CoV-2- exposed individuals, leading to lower estimated effectiveness.

While three doses of the COVID-19 vaccine demonstrated little protection against SARS-CoV-2 infections with the Omicron variant, four- and five- dose regimens showed moderately high degrees of protection (over 70%), which is consistent with Chariyalertsak’s study [18]. However, our result is different to Grewal et al., Bar-On et al., and Regev-Yochay et al.’s studies that reported the effectiveness of the fourth dose to be approximately 10% to 65% against Omicron variant infection [21,22,23]. This might be owing to the differing study’s population, as Grewal’s study was conducted among long-term care residents aged 60 years and over. In addition, the Thai government began rolling out the fourth dose in late January 2022. At the time of analysis, the follow-up time for this study was quite short; the mean time of the fourth and fifth dose follow-up period was only about two months. Therefore, the estimated vaccine effectiveness of fourth and fifth doses might be higher than in other studies. To assess the effectiveness of the fourth and fifth doses against infection, additional follow up is required.

Although the protective effect of three doses of the COVID-19 vaccine against any infection of SARS-CoV-2 in this study was low, the vaccine effectiveness of at least a two-dose regimen against severe pneumonia and deaths was obviously high. Our result is similar to a previous report from the UK which showed that two and three doses of the COVID-19 vaccine obviously provided protective benefits against hospitalization requiring oxygen support or intensive care, and the duration of effectiveness can be prolonged to more than 175 days during the Omicron epidemic [24]. Fourth dose effectiveness is also consistent with the study of Grewal et al. which demonstrated high protectiveness against severe outcomes [21]. The waning of effectiveness was minimal during the study period; however, to assess the waning of the vaccine, more follow-up time is needed.

In our study, the third dose and fourth dose of COVID-19 vaccine offer a high level of protection against severe outcome in the elderly. The finding is consistent with studies by Grewal et al. and Arregocés-Castillo et al. that demonstrated a high vaccine effectiveness against hospitalization and fatalities [21,25]. The results support the government policy on the use of booster doses (third dose and fourth dose) for the elderly, which is a vulnerable population.

In terms of severe outcomes, the vaccine effectiveness for those receiving two-dose mixing regimens was comparable to those acquitting homologous mRNA vaccines. All three-dose regimens provided a very high protective effect (80% to nearly 100%). The results are compatible with a previous study by Wing Ying Au et al. which found that heterologous regimens exhibited comparable effectiveness to homologous mRNA regimens [26]. However, some regimens such as two-dose BBIBP-CorV, might be confounded by the fact that these regimens were mostly offered to young and healthy populations leading to the possibility of overestimated value. In contrast, ChAdOx1 nCoV-19 plus ChAdOx1 nCoV-19 were widely provided to the elderly population and this might be one of the reasons to explain lower effectiveness of ChAdOx1 nCoV-19 plus ChAdOx1 nCoV-19, relative to two-dose BBIBP-CorV.

Our findings suggested that the current regimens of three doses of COVID-19 vaccines, homologous and heterologous, showed a high degree of protection against severe outcomes. This result supports the current national vaccine strategy of Thailand that ratifies the use of heterologous regimens. The booster dose of COVID-19 should be considered and administered in addition.

### Limitations

The study contains a few limitations. As the study used routine secondary data collected from various sources, the collected variables were limited, and the quality of data varied. This made the analysis prone to residual confounding and information bias. Information on some potential confounders such as underlying disease and detailed risk behavior were not available due to data inaccessibility. However, we attempted to limit the extent of confounding by including age in the modelling process. Age can be considered to be a proxy of underlying disease. Despite the matching on age to reduce the confounding effect of underlying diseases, we also limited the participants to those with the testing indications of contact with COVID-19 cases or being at risk of contracting the disease, which could be considered as a proxy of risk behavior. Due to a limited sample, the effectiveness of some vaccination regimens and doses could not be evaluated. Especially the time interval stratification results, we could not estimate effectiveness by a shorter interval, less than a three-month interval. The cases in the study included people who had a SARS-CoV-2 detection by either RT-PCR or ATK test. A person with ATK negative results might be actually a case (false negative) due to ineffective sample collection, ability of the test and inaccurate reporting, resulting in the misclassification of the outcome. This error from the sample collection is limited as samples were collected by professional health workers; however, further misclassification from other causes could not be avoided. For infection outcomes, misclassification of the outcome could not be evaded. This might be due to the test performance itself (self-ATK testing) and that the current reporting system in Thailand does not require all individuals with positive self-test ATK to report to the authorities. This meant the results of the self-test ATK were not mandatorily reported or linked with the Co-Lab surveillance. Then, some cases might be misclassified as control. Therefore, the effectiveness of COVID-19 infection from the study should be interpreted with caution. For severity outcomes, pneumonic cases needing invasive ventilation support were less likely to face misclassification of outcomes as the protocol of most hospitals in Thailand requires a patient with a severe condition to undertake RT-PCR before being admitted as an inpatient. For this reason, we decided to regard a severe condition as the primary outcome of this study. Note that non-differential misclassifications with the direction from non-severe to severe might still exist due to a lack of updated data on clinical severity after admission. In addition, as the study was conducted during the widespread of COVID-19 and could capture only those who tested positive for COVID-19 at hospitals, some samples may already have the protective immunity from infection prior to receiving the COVID-19 test at hospitals. This situation may diminish the estimated effectiveness of the COVID-19 vaccine. The waning of vaccine effectiveness could not be excluded as most of the samples were tested within five months after vaccine administration, especially the fourth and fifth doses which had only around two months of time interval between sample collection date and vaccination date. The results of the overall COVID-19 effectiveness, not stratified by vaccine regimen, have limited generalization, and should be interpreted with caution because it was based on a mix of the COVID-19 regimens that may differ to other countries. However, the effectiveness of some specific regimen combinations demonstrated in this study could be useful for other countries’ consideration. Data on individual-level variants was not available for all cases. We thus could not estimate the effectiveness of variant-specific outcomes. Instead, we relied on Thailand’s national variant coverage summary data to reaffirm that, at the time of the analysis, the Omicron variant was the most common circulating strain in the country. Lastly, as the study was conducted using a test-negative case–control study design which faced several limitations by itself, including the inability to estimate the actual infection rate of infection over time from the perspective of vaccination status. Thus, a cohort study or other study designs should be further conducted to identify the rate of infection and compare with the result of this study.

## 5. Conclusions

Despite our results that two and three doses of COVID-19 vaccination did not provide enough protection against any infection, the vaccines provided favourable effectiveness against severe pneumonia. The protective effect was even more notable among the three-dose vaccinees. No variation in vaccine effectiveness was detected across different age groups, and there was no remarkable decline in effectiveness against severe outcomes, at least within 180 days. The combination of vaccine regimens yielded satisfactory protection against serious illness. The Thai government should recommend its population receives the booster shot (third dose and onward). Monitoring the effectiveness of various vaccine regimens while accounting for the advent of any emerging SARS-CoV-2 variants and the waning time of vaccine protectiveness using cohort study or other study designs is recommended for sub-category analyses, particularly for the regimens with small numbers.

## Figures and Tables

**Figure 1 vaccines-10-02123-f001:**
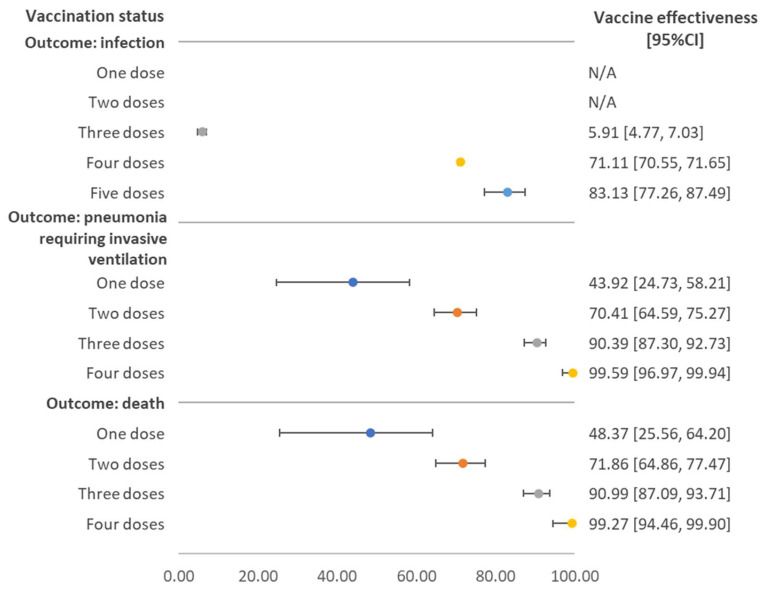
Vaccine effectiveness against SARS-CoV-2 infection, pneumonia requiring invasive ventilation, and death stratified by number of doses received, regardless of vaccine brand. Note: N/A referred to the fact that estimated value and its 95% confidence interval (CI) were in the negative form.

**Figure 2 vaccines-10-02123-f002:**
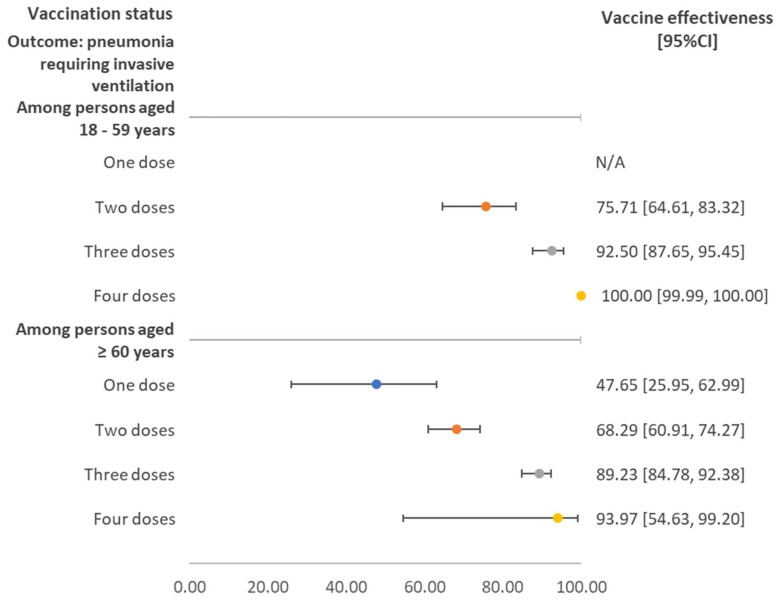
Vaccine effectiveness against pneumonia requiring invasive ventilation stratified by age group and number of doses received, regardless of vaccine brand. Note: N/A referred to the situation where the estimated value was insignificant, as the 95% confidence interval (CI) covered the null value.

**Figure 3 vaccines-10-02123-f003:**
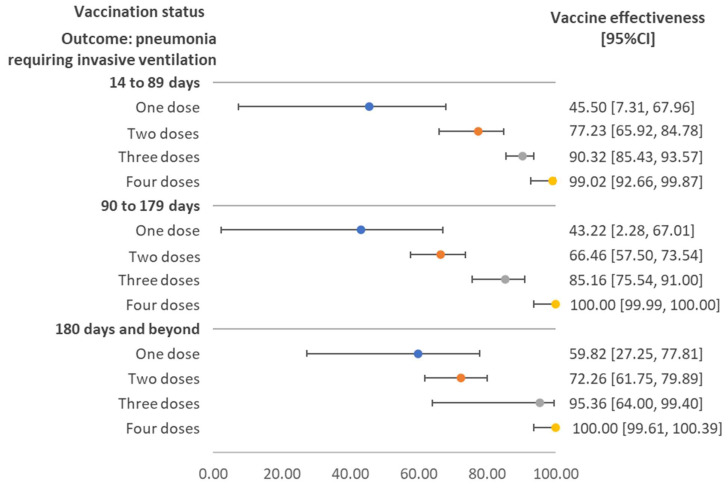
Vaccine effectiveness against pneumonia requiring invasive ventilation stratified by time interval between sample collection date and vaccination date, regardless of vaccine brand. Note: The maximum time interval (days) of cases with pneumonia requiring invasive ventilation and the compared group were 323 and 391 for one dose; 289 and 398 for two doses; 196 and 288 for three doses; and 42 and 255 for four doses. Only one pneumonia case requiring invasive ventilation had received four doses of vaccine. CI: confidence interval.

**Table 1 vaccines-10-02123-t001:** Characteristics of COVID-19 cases and controls.

Characteristics	Cases Sorted by Severity Status—n (%) (N = 652,854)			Non-SARS-CoV-2 Infection—n (%) (N = 2,406,762)
	SARS-CoV-2 Infection ^#^	Pneumonia Requiring Invasive Ventilation ^†^	Death	
	(N = 652,854)	(N = 1015)	(N = 656)	
Age (years) *	37.09 (20.26)	66.62 (18.26)	70.00 (15.55)	36.84 (19.91)
<12	73,224 (11.22)	20 (1.97)	1 (0.15)	259,587 (10.79)
12–17	53,678 (8.22)	4 (0.39)	0 (0.00)	211,591 (8.79)
18–59	425,793 (65.22)	284 (27.98)	167 (25.46)	1,586,611 (65.93)
60–120	100,123 (15.34)	707 (69.66)	488 (74.39)	348,875 (14.5)
Month of sample collection				
January	37,131 (5.69)	75 (7.39)	57 (8.69)	139,547 (5.80)
February	124,444 (19.06)	138 (13.60)	96 (14.63)	470,331 (19.54)
March	296,249 (45.38)	388 (38.23)	236 (35.98)	1,101,506 (45.77)
April	195,030 (29.87)	414 (40.79)	267 (40.7)	695,378 (28.89)
Vaccine doses received				
None	101,243 (15.51)	476 (46.90)	328 (50.00)	379,985 (15.79)
One	33,147 (5.08)	80 (7.88)	50 (7.62)	115,926 (4.82)
Two	335,498 (51.39)	376 (37.04)	231 (35.21)	1,071,713 (44.53)
Three	163,273 (25.01)	82 (8.08)	46 (7.01)	631,189 (26.23)
Four	19,646 (3.01)	1 (0.10)	1 (0.15)	207,194 (8.61)
Five	47 (0.01)	0 (0.00)	0 (0.00)	755 (0.03)
Time interval between last vaccination date and sample collection date (days) by vaccination doses received *				
One	85.15 (64.32)	130.58 (83.72)	118.78 (79.89)	86.10 (64.94)
Two	137.33 (47.48)	150.78 (51.31)	152.4 (50.5)	132.22 (48.22)
Three	80.91 (43.94)	75.94 (36.99)	80.98 (41.57)	79.58 (49.02)
Four	68.49 (28.2)	42.00 (0.00)	42.00 (0.00)	68.56 (27.25)
Five	53.83 (32.06)	N/A	N/A	55.29 (30.58)

Note: N/A: data were not available. ^#^ Total SARS-CoV-2 Infection included severe and non-severe COVID-19 cases. ^†^ Pneumonia requiring invasive ventilation included deaths. * Mean with standard deviation was presented.

**Table 2 vaccines-10-02123-t002:** Vaccine effectiveness against pneumonia requiring invasive ventilation for two-, three- and four-dose vaccination regimens.

Vaccine	Control	Pneumonia Requiring Invasive Ventilation ^†^	% VE	95%CI
Two-dose regimen				
Homologous vaccine regimen				
CoronaVac + CoronaVac	10,376	1	N/A	N/A
BBIBP-CorV + BBIBP-CorV	126,908	44	65.81	39.47–80.69
ChAdOx1 nCoV-19 + ChAdOx1 nCoV-19	176,771	85	58.41	39.95–71.19
BNT162b2 + BNT162b2	371,353	40	71.74	49.71–84.12
mRNA1273 + mRNA1273	16,386	4	N/A	N/A
Heterologous vaccine regimen				
CoronaVac + ChAdOx1 nCoV-19	578,790	176	71.70	63.22–78.22
CoronaVac + BNT162b2	34,374	4	79.79	7.36–95.59
ChAdOx1 nCoV-19 + BNT162b2	76,469	20	83.18	62.37–92.48
Three-dose regimen				
Two homologous vaccine regimens plus third vaccine				
CoronaVac + CoronaVac + ChAdOx1 nCoV-19	85,147	5	94.92	61.91–99.32
CoronaVac + CoronaVac + BNT162b2	51,522	0	100.00	99.99–100.00
BBIBP-CorV + BBIBP-CorV + BNT162b2	75,184	5	81.37	36.08–94.57
ChAdOx1 nCoV-19 + ChAdOx1 nCoV-19 + BNT162b2	166,083	21	85.84	74.00–92.29
ChAdOx1 nCoV-19 + ChAdOx1 nCoV-19 + mRNA1273	36,397	3	87.69	46.6–97.16
Two heterologous vaccine regimens plus third vaccine				
CoronaVac + ChAdOx1 nCoV-19 + ChAdOx1 nCoV-19	80,243	16	88.24	66.80–95.83
CoronaVac + ChAdOx1 nCoV-19 + BNT162b2	189,442	27	86.05	70.67–93.37
CoronaVac + ChAdOx1 nCoV-19 + mRNA1273	25,665	2	N/A	N/A
CoronaVac + BNT162b2 + BNT162b2	4383	0	100.00	99.99–100.00
ChAdOx1 nCoV-19 + BNT162b2 + BNT162b2	7733	1	N/A	N/A
ChAdOx1 nCoV-19 + BNT162b2 + mRNA1273	1062	0	100.00	99.99–100.00
Any two vaccines plus third vaccine				
Any two doses + ChAdOx1 nCoV-19	190,738	21	92.19	80.51–96.87
Any two doses + mRNA vaccine *	602,659	61	87.08	80.52–91.43
Four-dose regimen				
CoronaVac + CoronaVac + ChAdOx1 nCoV-19 + ChAdOx1 nCoV-19	9119	0	100.00	99.99–100.00
CoronaVac + CoronaVac + ChAdOx1 nCoV-19 + BNT162b2	65,682	0	100.00	99.99–100.00
CoronaVac + CoronaVac + ChAdOx1 nCoV-19 + mRNA1273	25,601	1	N/A	N/A
CoronaVac + CoronaVac + BNT162b2 + BNT162b2	100,971	0	100.00	99.99–100.00
CoronaVac + CoronaVac + mRNA1273 + mRNA1273	1275	0	100.00	99.99–100.00

Note: N/A shown that estimated value (VE) is insignificant or in the negative form. * The mRNA vaccines included BNT162b2and mRNA1273. ^†^ Pneumonia requiring invasive ventilation included deaths. CI: confidence interval.

## Data Availability

Not applicable.

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
