# Peer review of "Real-World Effectiveness of COVID-19 Vaccines against Severe Outcomes during the Period of Omicron Predominance in Thailand: A Test-Negative Nationwide Case–Control Study"

_vaccines, 2022, doi:10.3390/vaccines10122123_

Round 1
Reviewer 1 Report
The manuscript represents a general analysis of SARS CoV-2 vaccine schemes in Thailand. In general, the number of individuals analyzed is reasonable, but there are a few statistical type 2 errors that the authors partially acknowledge in the study's limitations. The mixture of combinations has huge limitations. The only valid general conclusion is that an increase in vaccine use decreases the severity of Covid19 assessed by oxygen therapy. Figures 1 to 3 represent an important bias and should be avoided. The bias refers to effectiveness, and by only monitoring one parameter is not appropriate. The rate of infection in non-vaccinated individuals was not compared. The discussion lacks of important points. The limitations are important.
Author Response
Thanks for your comments and suggestions. Figures 1 and 3 were also important as figure 1 demonstrated the level low of effectiveness against infection and figure 3 showed the decline of effectiveness across the study’s time. We agree that information bias could not be avoided. We added further detail about it in the limitation part.
“The cases in the study included people who had SARS-CoV-2 detection by either RT-PCR or ATK test. A person with ATK negative results might be actually a case (false negative) due to the ineffective sample collection, the ability of the test and the inaccurate reporting, resulting in the misclassification of the outcome. This type II error from the sample collection is limited as the samples were collected by professional health workers; however, further misclassification from other causes could not be avoided.”
Reviewer 2 Report
Authors have taken up the study on assessing the real-world effectiveness of COVID-19 vaccines against severe outcomes during when the predominance in Thailand. They used secondary data for a test-negative nationwide case-control study. The authors claimed that the study reveals the trajectory of the real-world effectiveness of COVID-19 vaccines and concluded the current national vaccine policies are potentially effective against the Omicron variant in Thailand.
The main advantage is that the authors used nationwide data for a test-negative nationwide case-control study. But the manuscript has several scopes of improvement, including the reliable claim and formation of research questions. I anticipate authors to take up the following issues to improve the manuscript.
1. My prime concern is on the claim of the ‘trajectory’ of the real-world effectiveness of COVID-19 vaccines. In fact, I am unable to any results on the trajectory in particular. The results are based on secondary data and retrospectively designed for the test-negative case-control study. I suggest authors to clarify their claim or revise it accordingly.
2. In connection to the above issue, I wonder authors could have parallelly analyzed the data at a temporal scale, to examine the timely trajectory of the effectiveness of different vaccines and outcomes (against which the effectiveness is calculated).
3. Further, the title is to be as per the results presented. Based on the responses to the above issues the title should be revised in terms of using ‘trajectory’ terms.
“Trajectory of real-world effectiveness of COVID-19 vaccines against severe outcomes during the period of Omicron predominance in Thailand: a test-negative nationwide case-control study”.
4. In the abstract: “A conditional logistic regression was performed.” Revise the sentence with linked to the objective of such analysis. Suggest not to have such isolated sentences in the abstract.
5. Figures 1-3 could be improved in the presentation. Do we really need three separate figures for it? What is the reason to provide both estimates and graphical illustrations together? It seems a table is a better presentation for it. Authors should combine all three figures in one with very clear scales for effectiveness and can use different colors for outcomes. So that the figure should illustrate the comparison in a single figure efficiently. Yes, the estimates can be put as a table and kept in the appendix, and discussed in the text.
6. Further, I would prefer to keep the estimates instead of saying ‘NA’ for the insignificant estimates. The authors should mention the potential reasons for having insignificant/negative effectiveness. Is it due to the sample size issue? Avoid using ‘NA’ without a clear reason, if it is insignificant reader will understand looking at the estimate, but the authors should explain such results as well.
7. “… As Thailand is in the transition of COVID-19 as an emerging disease to an endemic disease, and the national strategy …”. Suggest revising the sentence as COVID-19 is going to be endemic in Thailand only but also worldwide.
8. “The study population was Thai individuals tested for SARS-CoV-2 between January and April 2022 by healthcare professionals.” Suggest to provide the exact date here along with the months.
9. Authors have found two and three doses of COVID-19 vaccination did not provide enough protection against any infection; the vaccine provided favorable effectiveness against severe pneumonia. I suggest authors could discuss this in context with recently published evidence-based studies (e.g., Cheng S et al Nat Med 2022).
The concluding paragraph can be improved by making it data-driven in specific instead general comparison and advancement of the framework. The present form is not in the context of the research question. Finally, add the implication of these outcomes.
Author Response
Reviewer 2
Authors have taken up the study on assessing the real-world effectiveness of COVID-19 vaccines against severe outcomes during when the predominance in Thailand. They used secondary data for a test-negative nationwide case-control study. The authors claimed that the study reveals the trajectory of the real-world effectiveness of COVID-19 vaccines and concluded the current national vaccine policies are potentially effective against the Omicron variant in Thailand.
The main advantage is that the authors used nationwide data for a test-negative nationwide case-control study. But the manuscript has several scopes of improvement, including the reliable claim and formation of research questions. I anticipate authors to take up the following issues to improve the manuscript.
- My prime concern is on the claim of the ‘trajectory’ of the real-world effectiveness of COVID-19 vaccines. In fact, I am unable to any results on the trajectory in particular. The results are based on secondary data and retrospectively designed for the test-negative case-control study. I suggest authors to clarify their claim or revise it accordingly.
Answer Thanks for your comment. We used the word “trajectory” to represent the estimated VE during the specific time period after vaccine administration over the six-month course, as shown in Figure 3. I agree with your comment that this term may not be suitable for this analysis, so I remove it from the article.
- In connection to the above issue, I wonder authors could have parallelly analyzed the data at a temporal scale, to examine the timely trajectory of the effectiveness of different vaccines and outcomes (against which the effectiveness is calculated).
Answer Thanks for your comment. It is an interested one. We only examined the vaccine effectiveness of each dose of vaccine at a temporal scale as shown in Figure 3. We could not stratify the analysis by regimens as the sample size for each regimen is limited.
- Further, the title is to be as per the results presented. Based on the responses to the above issues the title should be revised in terms of using ‘trajectory’ terms.
“Trajectory of real-world effectiveness of COVID-19 vaccines against severe outcomes during the period of Omicron predominance in Thailand: a test-negative nationwide case-control study”.
Answer We removed the term “trajectory” from the title of the article.
- In the abstract: “A conditional logistic regression was performed.” Revise the sentence with linked to the objective of such analysis. Suggest not to have such isolated sentences in the abstract.
Answer We revised the sentenced to “The case and control were matched with the ratio of 1:4 in terms of age, date of specimen collection and hospital collection specimen and odds ratio was calculated using conditional logistic regression.
- Figures 1-3 could be improved in the presentation. Do we really need three separate figures for it? What is the reason to provide both estimates and graphical illustrations together? It seems a table is a better presentation for it. Authors should combine all three figures in one with very clear scales for effectiveness and can use different colors for outcomes. So that the figure should illustrate the comparison in a single figure efficiently. Yes, the estimates can be put as a table and kept in the appendix, and discussed in the text.
Answer Thanks for your suggestion. We try to change the presentation to table; however, it is difficult to capture all of the information. The estimated value outside the graph was presented to let the audience knew the actual value. So, we would like to keep the current format of the results’ presentation.
- Further, I would prefer to keep the estimates instead of saying ‘NA’ for the insignificant estimates. The authors should mention the potential reasons for having insignificant/negative effectiveness. Is it due to the sample size issue? Avoid using ‘NA’ without a clear reason, if it is insignificant reader will understand looking at the estimate, but the authors should explain such results as well.
Answer Thanks for your comment. We add the specific reason of NA in the footnote.
In figure 1, We changed the description to “N/A referred to the fact that estimated value and its 95% confidence interval were in the negative form.”
In figure 2, we changed to “N/A referred to the situation that the estimated value was insignificant as the 95% confidence interval covered the null value.”
- “… As Thailand is in the transition of COVID-19 as an emerging disease to an endemic disease, and the national strategy …”. Suggest revising the sentence as COVID-19 is going to be endemic in Thailand only but also worldwide.
Answer Thanks for your suggestion. We revised the sentence per your suggestion to “As COVID-19 is going to be an endemic disease not only in Thailand but also worldwide, this study paid attention to the severe outcomes of the disease rather than overall number of infectees”.
- “The study population was Thai individuals tested for SARS-CoV-2 between January and April 2022 by healthcare professionals.” Suggest to provide the exact date here along with the months.
Answer Thanks for your comment. We revised it and added the date to the method part.
- Authors have found two and three doses of COVID-19 vaccination did not provide enough protection against any infection; the vaccine provided favorable effectiveness against severe pneumonia. I suggest authors could discuss this in context with recently published evidence-based studies (e.g., Cheng S et al Nat Med 2022).
Answer Thanks for your suggestion. We further discussed this point and added an updated papers for references.
“Compared with the neutralizing antibody study by Cheng and his team, this study showed different results. Cheng’s study suggested that the three doses of COVID-19 vaccine increased the neutralizing antibody levels after vaccine administration. This might be caused by the difference in the time period of interest, as the study employed longer study duration after the third doses.”
- The concluding paragraph can be improved by making it data-driven in specific instead general comparison and advancement of the framework. The present form is not in the context of the research question. Finally, add the implication of these outcomes.
Answer Thanks for your comments. We clarified about the implication of the outcome.
“Our findings suggested that the current regimens of three doses of COVID-19 vaccines, homologous and heterologous, showed a high degree of protection against severe outcome. This result supports the current national vaccine strategy of Thailand that ratifies the use of heterologous regimens. The booster dose of COVID-19 should be considered and administered.”
Reviewer 3 Report
Dear Authors,
the following are the observations:
1) Why you write "real-world" in the title and in the text?
2) Have you considered the comorbidities of the subjects in your study? Why these aren't present in the article? (Expecially when you explain the figure 2 and 3);
3) Why did you not consider the cases of positivity detectable by blood sampling?
4) The table 1: row "Age" has 5 values for column but the range of the age is of 4 elements;
5) As wrote by authors of: "Persistence and performance of memory B cells in vaccinated health care workers with breakthrough infections"
https://www.sciencedirect.com/science/article/pii/S1931312822000397
"Highly specific memory B cells generation after 2nd dose of BNT162b2 vaccine compensate for the decline of serum antibodies and absence of mucosal IgA"
https://www.mdpi.com/2073-4409/10/10/2541
the second dose has an its effectiveness agaist the effects of the infection
Author Response
Reviewer 3
the following are the observations:
1) Why you write "real-world" in the title and in the text?
Answer Because the study is conducted based on the secondary data which were acquired under normal circumstances in the real-world setting.
2) Have you considered the comorbidities of the subjects in your study? Why these aren't present in the article? (Especially when you explain the figure 2 and 3);
Answer Thanks for your comments. We try to identify the comorbidity of the subjects; however, we cannot access the database in which collected these data. It is one of our limitations. We revised the limitation by adding the reason in the sentence “Information on some potential confounders such as underlying disease and detailed risk behavior were not available due to data inaccessibility.”
3) Why did you not consider the cases of positivity detectable by blood sampling?
Answer In Thailand, the suspected COVID-19 patients were tested by RT-PCR or ATK using nasopharyngeal swab or sputum. We did not collect the blood sampling for confirmation of COVID-19.
4) The table 1: row "Age" has 5 values for column, but the range of the age is of 4 elements;
Answer The first row represented the mean with standard deviation of age, while the second to the fifth row showed the frequency and percentage of subjects in each category.
5) As wrote by authors of: "Persistence and performance of memory B cells in vaccinated health care workers with breakthrough infections"
https://www.sciencedirect.com/science/article/pii/S1931312822000397
"Highly specific memory B cells generation after 2nd dose of BNT162b2 vaccine compensate for the decline of serum antibodies and absence of mucosal IgA"
https://www.mdpi.com/2073-4409/10/10/2541
the second dose has an its effectiveness against the effects of the infection
Answer Thanks for your comment. The disparity of the results might be due to the difference in the study design, vaccine type, duration, and study’s population as the above paper was mainly focused on the immunity level while the study used the field data.
Round 2
Reviewer 1 Report
The changes in the manuscript were very few. The queries were not properly addressed. There are still issues with the grammar which weren't solved.
Reviewer 3 Report
Dear Authors,
can you add this point in the abstract, near the word "real"? the study is conducted based on the secondary data which were acquired under normal circumstances in the real-world setting.
Best regards.
Author Response
Thanks for your comment. We add the description of the real world in the sentence.
“The primary objective of this study is to examine the real-world effectiveness of COVID-19 vaccines based on secondary data acquired under normal circumstances in a real-world setting, to protect against treatment with invasive ventilation of pneumonia during January to April 2022, a period when omicron was predominant."
Round 3
Reviewer 1 Report
The manuscript has been improved even though the authors did not respond to all the comments. The manuscript can be published.